# Compartment-specific opioid receptor signaling is selectively modulated by different dynorphin peptides

Jennifer M Kunselman[1†], Achla Gupta[2†], Ivone Gomes[2], Lakshmi A Devi[2], Manojkumar A Puthenveedu[1,3*]

[1]Cellular and Molecular Biology Training Program, University of Michigan Medical School, Ann Arbor, United States; [2]Department of Pharmacological Sciences, Icahn School of Medicine at Mount Sinai, New York, United States; [3]Department of Pharmacology, University of Michigan Medical School, Ann Arbor, United States

**Abstract** Many signal transduction systems have an apparent redundancy built into them, where multiple physiological agonists activate the same receptors. Whether this is true redundancy, or whether this provides an as-yet unrecognized specificity in downstream signaling, is not well understood. We address this question using the kappa opioid receptor (KOR), a physiologically relevant G protein-coupled receptor (GPCR) that is activated by multiple members of the Dynorphin family of opioid peptides. We show that two related peptides, Dynorphin A and Dynorphin B, bind and activate KOR to similar extents in mammalian neuroendocrine cells and rat striatal neurons, but localize KOR to distinct intracellular compartments and drive different post-endocytic fates of the receptor. Strikingly, localization of KOR to the degradative pathway by Dynorphin A induces sustained KOR signaling from these compartments. Our results suggest that seemingly redundant endogenous peptides can fine-tune signaling by regulating the spatiotemporal profile of KOR signaling.

*For correspondence: puthenve@umich.edu

†These authors contributed equally to this work

Competing interests: The authors declare that no competing interests exist.

## Introduction

The endogenous opioid system provides an excellent and physiologically relevant example to study redundancy in signaling systems in our body. Over 20 endogenous opioids have been identified, all of which preferentially activate one of three opioid receptors – delta, kappa, and mu opioid receptors – which are all members of the G protein-coupled receptor (GPCR) family of proteins (*Gendron et al., 2016*; *Chavkin, 2013*; *Williams et al., 2013*). All these opioid peptides activate their cognate GPCRs broadly at similar levels in most of the readouts that have been used to measure activation (*Sternini et al., 2013*). Whether all these opioid peptides are truly redundant or whether they contribute to signaling diversity beyond the initial signaling has been a long-standing question in the field.

Signaling from intracellular compartments, after the initial signaling from the surface, is emerging as a key determinant of the downstream consequences of receptor activation (*Irannejad et al., 2013*; *Yarwood et al., 2017*; *Eichel and von Zastrow, 2018*). While this is still an emerging field, a growing body of evidence suggests that GPCRs are active in endosomes and other intracellular compartments, and that receptors in endosomes can cause distinct signaling outcomes compared to receptors on the cell surface (*Vilardaga et al., 2014*; *Bowman et al., 2016*; *Stoeber et al., 2018*). Receptors rapidly and dynamically move between intracellular compartments and the surface by trafficking. Trafficking could therefore act as a master regulator of GPCR signaling by selectively amplifying signals from specific locations (*Weinberg et al., 2019*; *Hanyaloglu, 2018*). Whether

physiological systems take advantage of trafficking to localize receptors to different compartments and dictate location-biased signaling, however, is still unanswered.

Here we bridge both questions by asking whether different endogenous opioid peptides can sort receptors to distinct intracellular compartments and drive different location-biased signaling outcomes. Using the kappa opioid receptor (KOR) as a model GPCR, we show that although related Dynorphin peptides can activate KOR on the surface to a similar extent, they induce different trafficking fates and endosomal localization of the receptor. Dyn B predominantly localized KOR to Rab11 recycling endosomes and caused KOR recycling, while Dyn A predominantly localized KOR to late endosomes and caused KOR degradation. Strikingly, Dyn A-activated KOR, but not Dyn B-activated KOR, was in an active conformation in lysosomes and induced cAMP signaling from intracellular compartments. The differences are likely a result of differences in how the peptides activate KOR, as opposed to peptide stability. Our results show that seemingly redundant opioid peptides, which activate receptors on the surface to similar levels, can drive spatially and temporally different signaling outcomes by differentially sorting receptors to distinct endosomal compartments after internalization.

## Results

To examine activation and internalization of KOR by Dynorphin peptides we first focused on four physiologically relevant endogenous peptides – Dynorphin A17 (Dyn A), Dynorphin A8 (Dyn A8), Dynorphin B (Dyn B), and α-neoendorphin (α-neo). These peptides differ mainly in their length and C-terminal peptide sequence (*Figure 1A*). We carried out our studies using neuroendocrine PC-12 cells stably expressing KOR tagged with a pH-sensitive GFP (SpH-KOR) (*Sankaranarayanan et al., 2000*) that facilitates visualization of agonist-mediated KOR trafficking, and CHO cells stably expressing Flag epitope tagged KOR (CHO-KOR). Consistent with previous findings, we find that in these cells Dyn A, Dyn A8, Dyn B, and α-neo bind KOR at relatively comparable affinities (*Table 1*).

Next, we measured the inhibition of cAMP levels and KOR endocytosis induced by these peptides. Dose–response curves with Dyn A and Dyn B in SpH-KOR cells showed maximal inhibition ~1 µM (*Figure 1—figure supplement 1A*). At this concentration, Dyn A8 and α-neo inhibited whole-cell cAMP to levels comparable to that of Dyn A and Dyn B (*Figure 1B*). Measurement of cell surface levels of KOR by ELISA show that Dyn A and Dyn B endocytose the receptor to a similar extent in SpH-KOR cells and in CHO-KOR cells (*Figure 1—figure supplement 1B and C*; *Gupta et al., 2016*), with maximal endocytosis at ~1 µM. Examination of surface SpH-KOR fluorescence by live cell imaging using Total Internal Reflection Microscopy (TIR-FM) shows similar extent of agonist-mediated KOR clustering into endocytic domains and receptor endocytosis by the four Dynorphin peptides (1 µM) (*Figure 1C and D*). Together these results indicate that different Dynorphin peptides activate KOR and induce KOR internalization to similar levels.

We next examined if different Dynorphins selectively regulate the fate of KOR after initial activation and internalization, by adapting a discrete-event imaging method to quantitate the rate of individual KOR recycling events over unit time. SpH-KOR fluorescence is quenched in acidic endosomal compartments and is rapidly dequenched when receptors recycle back to the cell surface and are exposed to the extracellular media. This dequenching can be visualized as single events using Total Internal Reflection Fluorescence microscopy (TIR-FM), where the recycling events appear as distinctive sudden spikes in fluorescence followed by an exponential decay as the receptors diffuse on the cell membrane (*Figure 2A–C*). This method allows us to quantitate individual recycling events in the same cells over time without the confounding effects of continuing endocytosis (*Kunselman et al., 2019*). When the number of SpH-KOR recycling events were quantitated and normalized to time and cell area, a significantly higher number of recycling events was seen 5 min after Dyn B-induced KOR internalization , compared to Dyn A, Dyn A8, or α-neo (*Figure 2D*).

To test whether this increase in the rate of discrete recycling events corresponded to an increase in receptor levels at the cell surface, we measured ensemble changes in surface KOR levels by two different methods – whole-cell fluorescence and ELISA-based methods. We focused on Dyn A and Dyn B as a highly relevant and interesting pair, as both are processed from adjacent regions of prodynorphin and are often co-expressed in physiologically relevant brain regions (*Nikoshkov et al., 2005*; *Corder et al., 2018*). When SpH-KOR fluorescence was followed by live confocal imaging, surface fluorescence decreased after Dynorphin addition, as was expected with receptor internalization.

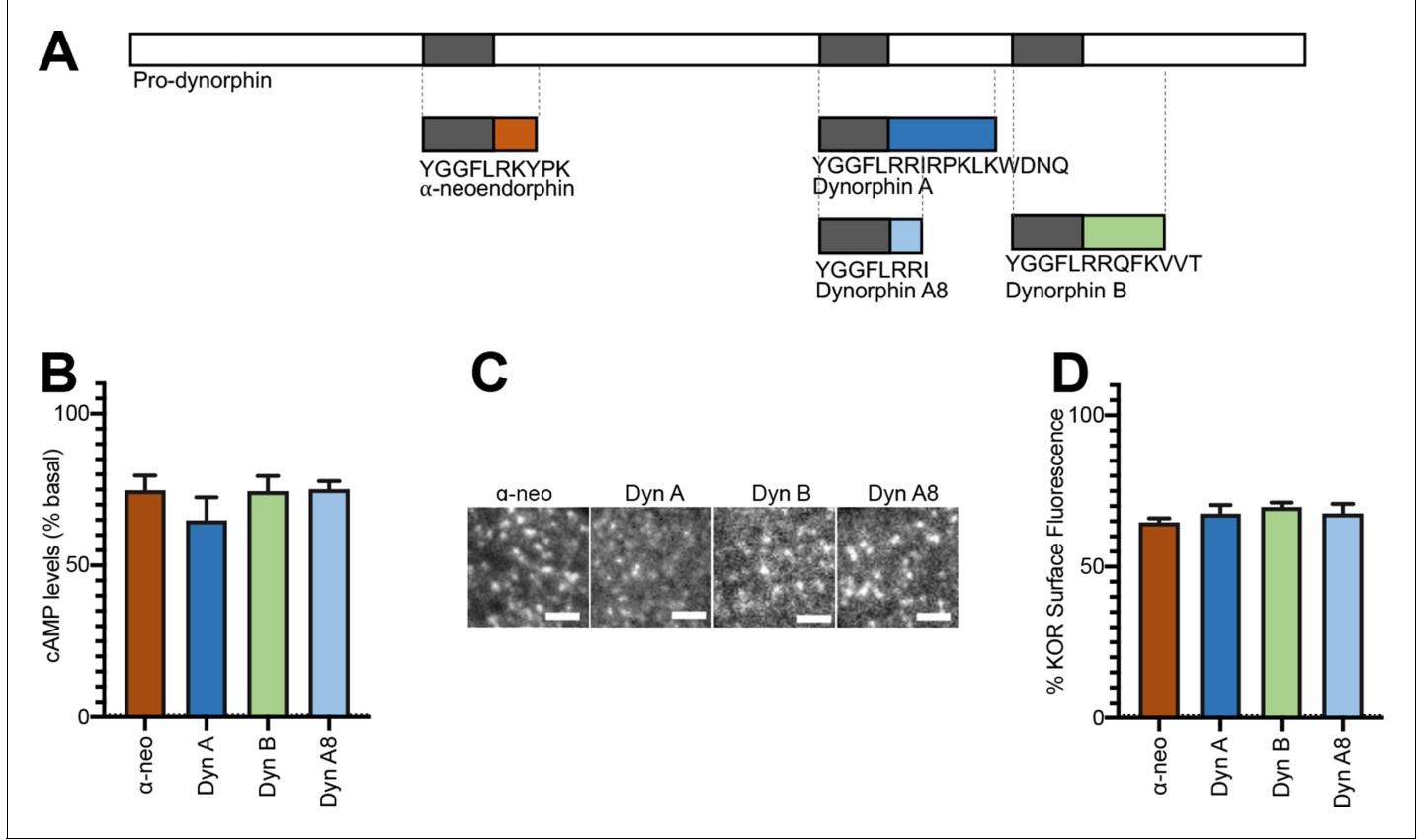

**Figure 1.** Initial activation and internalization of KOR by Dynorphins are comparable. (**A**) Schematic of the regions of Pro-dynorphin from which Dynorphin A8 (Dyn A8), Dynorphin A (Dyn A), Dynorphin B (Dyn B), and α-neoendorphin (α-neo) peptides are generated, showing that Dyn A and Dyn B are processed from adjacent regions. (**B**) Dyn A8, Dyn A, Dyn B, and α-neo (1 μM) inhibit intracellular cAMP levels to a similar extent in PC12 cells stably expressing SpH-KOR. Values were normalized to basal cAMP measurements in the absence of peptide, which were set as 100% (mean ± SEM shown). (**C**) Representative TIR-FM images of PC12 cells stably expressing SpH-KOR treated with Dyn A8, Dyn A, Dyn B, and α-neo (1 μM) show roughly similar agonist-mediated receptor clustering at the cell surface following 1 min treatment. Scale bar = 2 μm. (**D**) Quantitation of the loss of surface SpH-KOR fluorescence, as an index of internalization, after 5 min of treatment with each peptide (1 μM), normalized to surface fluorescence before agonist treatment, show similar levels of internalization for all four peptides (mean ± SEM shown, Dyn A: n = 10 cells; Dyn B: n = 10 cells; Dyn A8: n = 10 cells; α-neo: n = 11 cells).

The online version of this article includes the following figure supplement(s) for figure 1:

**Figure supplement 1.** Ligand-mediated decreases in intracellular cAMP levels and endocytosis of KOR saturates at 1 μM for Dyn A and Dyn B.

The fluorescence decrease reached a plateau at 10 min, suggesting an equilibrium between endocytosis and recycling at this time point (*Figure 2E*). When agonist was removed by washing out the media and replacing with fresh media containing antagonist (naltrexone; 10 μm), to specifically

**Table 1.** Displacement binding parameters for Dynorphin peptides at PC12 SPH-KOR and CHO-KOR cells.

| Ligand | PC12 SpH-KOR cells | | | | CHO-KOR cells | | | |
|---|---|---|---|---|---|---|---|---|
| | Low $K_i$ (nM) | High $K_i$ (nM) | % $B_{max}$ at 10 μM | $n_H$ | Low $K_i$ (nM) | High $K_i$ (nM) | % $B_{max}$ at 10 μM | $n_H$ |
| Dyn A8 | 56.9 ± 0.3 | 0.020 ± 1.90 | 18.30 ± 1.48 | 36.9% | 563 ± 0.1 | 0.41 ± 0.15 | 21.51 ± 1.54 | 41.6% |
| Dyn A17 (Dyn A) | 52.4 ± 0.3 | 0.016 ± 0.47 | 15.22 ± 1.56 | 39.1% | 119 ± 0.2 | 0.26 ± 0.54 | 18.72 ± 1.09 | 29.4% |
| Dyn B13 (Dyn B) | 37.8 ± 0.2 | 0.010 ± 0.31 | 11.13 ± 1.07 | 33.4% | 355 ± 0.1 | 0.38 ± 0.11 | 17.16 ± 0.82 | 42.2% |
| a-neo-endorphin | 39.1 ± 0.1 | 0.011 ± 0.20 | 13.32 ± 1.84 | 38.7% | 591 ± 0.1 | 0.18 ± 0.34 | 20.55 ± 1.57 | 21.4% |

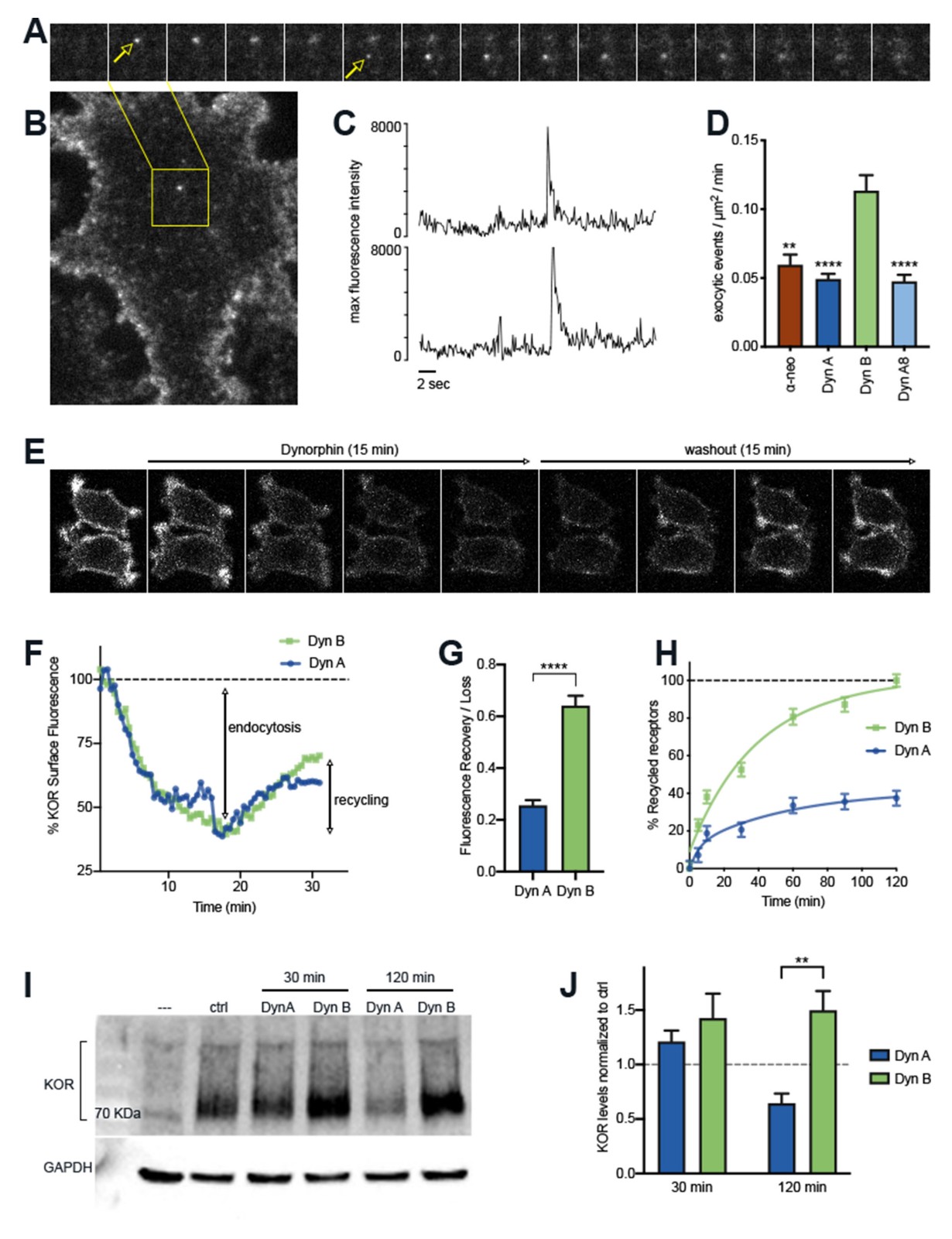

**Figure 2.** The post-endocytic fate of KOR is determined by the specific Dynorphin that activates it. (**A**) Frames from a time lapse movie of a representative region of a PC12 cell stably expressing SpH-KOR (SpH-KOR cells) shown in (**B**), treated for 5 min with Dyn B (1 μM) and imaged in TIR-FM, showing two examples of exocytic events (white arrows in **A**) associated with KOR recycling. (**C**) Fluorescence traces of the two exocytic events, arbitrary units, showing a characteristic abrupt increase in maximum fluorescence intensity followed by exponential decay. (**D**) Quantitation of the

*Figure 2 continued on next page*

Figure 2 continued

number of exocytic events/µm²/min showing a significant increase for Dyn B compared to the other peptides (mean ± SEM, **p<0.01, ****p<0.0001 in multiple comparisons after ANOVA, n = 14, 39, 52, and 33 cells for α-neo, Dyn A, Dyn B, and Dyn A8, respectively). (E) Ensemble SpH-KOR surface fluorescence measured over time using confocal microscopy shows a decrease in fluorescence upon agonist addition because of quenching of internalized SpH-KOR, and an increase upon peptide washout as receptors recycle to the surface and SpH-KOR is dequenched. (F) Quantification of change in ensemble surface fluorescence over 30 min following treatment with Dyn A or Dyn B (1 µM), normalized to fluorescence before agonist addition, showing the loss during endocytosis and increase during recycling. (G) Quantitation of the amount of SpH-KOR recycled, normalized to the amount endocytosed after Dyn A or Dyn B treatment B (1 µM), shows that a higher amount of receptor is recycled after Dyn B washout (mean ± SEM, ****p<0.0001 by Mann–Whitney; n = 33 and 30 fields for Dyn A and Dyn B, respectively). (H) Recycling of SpH-KOR to the cell surface after treatment with 100 nM Dyn A or Dyn B measured by ELISA shows a much higher rate and extent of recycling after Dyn B washout. (I) Representative immunoblot of total receptor levels in SpH-KOR cells treated with cycloheximide (3 µg/ml) for 2 hr prior to Dyn A or Dyn B treatment (1 µM) for the indicated times show receptor loss after 120 min of Dyn A but not Dyn B treatment. GAPDH is used as a control. (J) Quantification of total receptor levels normalized to untreated control cells under each condition (**p<0.01 by post hoc comparison after two-way ANOVA; n = 5).

The online version of this article includes the following figure supplement(s) for figure 2:

**Figure supplement 1.** The differences in KOR recycling between Dyn A and Dyn B cannot be explained entirely by peptide degradation.

measure recycling without the contribution of endocytosis, the fluorescence recovery rate was higher with Dyn B than with Dyn A at the 30 min time point (*Figure 2F and G*).

A potential reason for the differences we observe could be differences in the rate of proteolytic degradation of the peptides (*Mzhavia et al., 2003*; *Fricker et al., 2020*). Therefore, we directly tested whether inhibiting proteolysis abolishes the difference between Dyn A and Dyn B. We first used a protease inhibitor cocktail in the media to inhibit general proteolysis. When discrete recycling events were quantitated as in *Figure 2D*, Dyn B showed a higher number of KOR recycling events even in the presence of protease inhibitors (*Figure 2—figure supplement 1A*). When the recovery of surface KOR levels after agonist washout were measured by an independent ELISA-based method, Dyn B-treated cells showed a higher rate of recovery compared to Dyn A, even in the presence of protease inhibitors (*Figure 2H*). These experiments indicate that differences in general proteolysis of the peptides did not contribute to the increased KOR recycling we observe with Dyn B. Because Dyn B, but not Dyn A, can be cleaved by Endothelin Converting Enzyme 2 (ECE2) in vitro, we next tested whether this differential proteolysis by ECE2 could contributes to the differences between Dyn A and Dyn B-induced KOR recycling. We used S136492, which inhibits ECE2 relatively selectively in vitro in purified systems (*Mzhavia et al., 2003*), to test whether ECE2 activity was required for Dyn B to drive increased recycling. S136492 significantly reduced recycling for both Dyn A and Dyn B, indicating that the differences in trafficking between Dyn B and Dyn A cannot be explained solely by ECE2 sensitivity (*Figure 2—figure supplement 1B*). Together, these results suggest that intrinsic differences between Dyn A and Dyn B contribute to differences in KOR recycling when activated by these peptides.

Because KOR did not recycle efficiently when activated by Dyn A, we next asked whether Dyn A-activated KOR was sorted into the degradative pathway. To test this, PC12 cells expressing SpH-KOR were pretreated with cycloheximide (3 µg/ml) 2 hr before agonist addition to inhibit any new protein synthesis and to measure agonist-mediated turnover of KOR. Total KOR levels were determined through immunoblotting, after Dyn A or Dyn B treatment for 30 min or 2 hr (*Figure 2I*). When total receptor levels were quantified, 2 hr treatment with Dyn A caused a loss of 50% of KOR, while Dyn B treatment caused no loss (*Figure 2J*). These results suggest that, after endocytosis, Dyn A preferentially sorts KOR into the degradative pathway, while Dyn B sorts KOR into the recycling pathway.

Considering the emerging importance of spatial encoding in diversifying the outcomes of GPCR signaling (*Eichel and von Zastrow, 2018*; *Weinberg et al., 2019*; *Hanyaloglu, 2018*), we next asked whether Dyn A or Dyn B generated distinctive intracellular localization patterns of KOR at steady state. Because Dyn A drove KOR degradation, we first tested whether Dyn A-activated KOR was differentially localized to the late endosomal pathway. When PC12 cells expressing FLAG-KOR and Rab7-GFP, to mark the late endocytic pathway, were treated with 1 µM Dyn A for 20 min and imaged live, KOR colocalized predominantly with Rab7 (example in *Figure 3A*). To quantitate the distribution of KOR in the endosomal pathway more comprehensively, we treated cells expressing SpH-KOR with 1 µM Dyn A or Dyn B for 20 min, and fixed and stained for APPL1 (very early

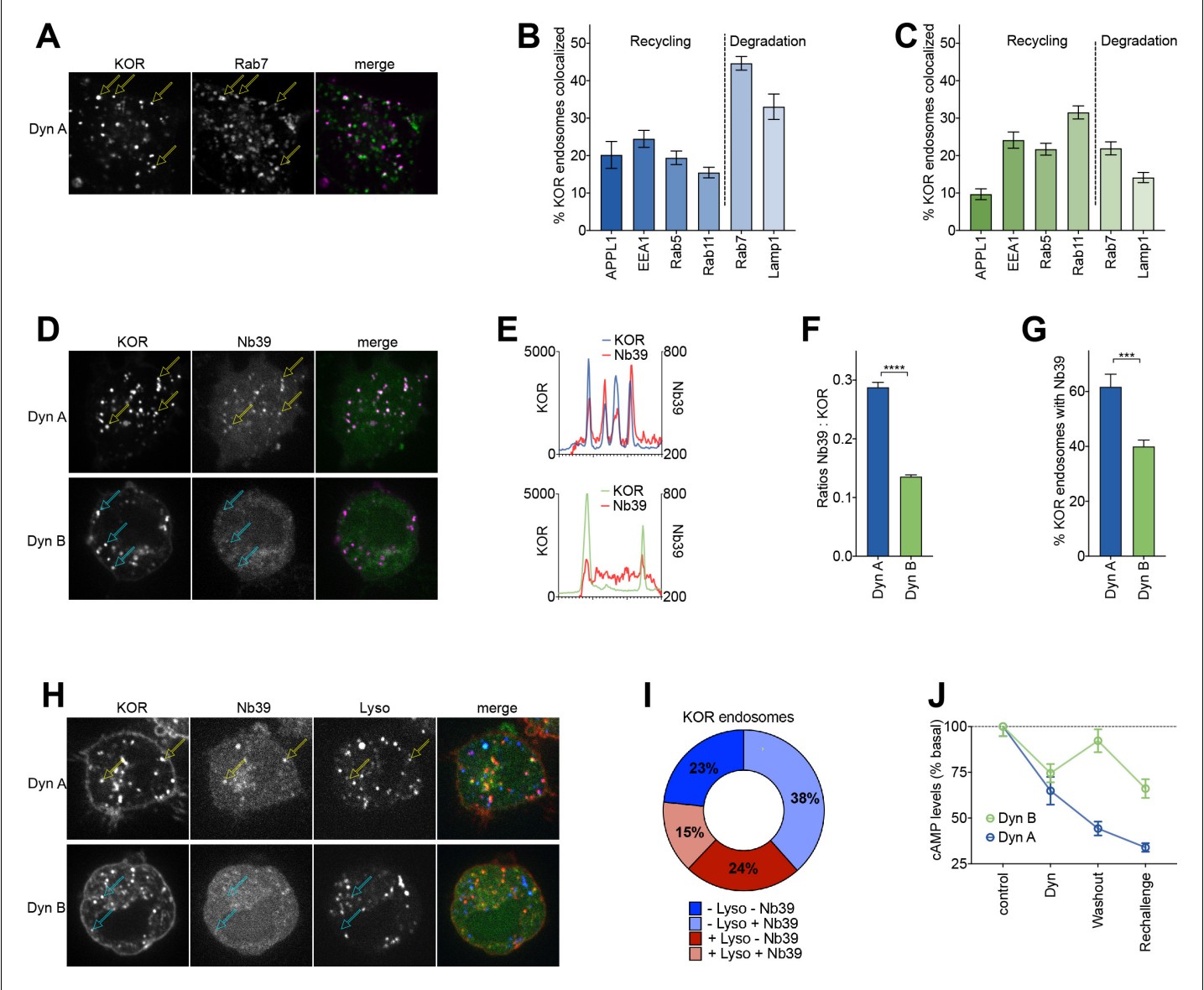

**Figure 3.** Dyn A selectively drives KOR signaling from late endosomal compartments. (**A**) Representative image of a PC12 cell expressing FLAG-KOR and Rab7-GFP, treated with 1 µM Dyn A for 20 min. Yellow arrows denote KOR endosomes that colocalize with Rab7. (**B**) SpH-KOR cells treated with 1 µM Dyn A for 20 min were fixed and processed for immunofluorescence with the noted markers. Quantitation, across multiple cells, of the percentage of KOR containing endosomes that colocalize with each of the endosomal markers is noted. KOR primarily localizes in Rab7 and Lamp1 positive late endosomes after Dyn A (n = 8, 10, 9, 11, 20, and 17 cells for APPL1, EEA1, Rab5, Rab11, Rab7, and Lamp1, respectively). (**C**) A similar quantitation of immunofluorescence images after Dyn B treatment (1 µM for 20 min) shows that KOR localizes less with late endosomes, and more with markers of early/recycling endosomes (n = 18, 16, 15, 18, 23, and 29 cells for APPL1, EEA1, Rab5, Rab11, Rab7, and Lamp1, respectively). (**D**) Representative images of PC12 cells expressing FLAG-KOR and Nb39, imaged live after treatment with 1 µM Dyn A or Dyn B for 20 min. Yellow arrows in Dyn A show KOR endosomes that recruited Nb39, while cyan arrows in Dyn B show KOR endosomes that do not show obvious recruitment of Nb39. (**E**) Linear profile plots of fluorescence of KOR and Nb39, measured along lines drawn across regions of the cell with KOR endosomes after treatment with 1 µM Dyn A or Dyn B for 20 min, show that Nb39 fluorescence increases along with KOR in Dyn A, but less noticeably with Dyn B. (**F**) Ratios of integrated fluorescence of Nb39:KOR in endosomes identified by 3D object analysis show higher amounts of Nb39 relative to KOR in Dyn A-treated cells (****p<0.0001 by Mann–Whitney, n = 766 and 800 endosomes for Dyn A and Dyn B, respectively). (**G**) Quantitation of the percentage of KOR endosomes per cell with a noticeable increase in Nb39 fluorescence above background shows a higher fraction of KOR endosomes recruited Nb39 in 1 µM Dyn A-treated cells (***p<0.001 by Mann–Whitney, n = 11 and 14 cells for Dyn A and Dyn B, respectively). (**H**) Representative images of PC12 cells expressing FLAG-KOR and Nb39 and labeled with LysoTracker imaged live after treatment with 1 µM Dyn A or Dyn B for 20 min. Yellow arrows in Dyn A show KOR endosomes that recruited Nb39 that were also labeled with Lysotracker, while cyan arrows in Dyn B show KOR endosomes that do not show obvious labeling with Nb39 and Lysotracker. (**I**) The average composition of total KOR endosomes that are positive for ±Nb39 and ±Lysotracker

*Figure 3 continued on next page*

*Figure 3 continued*

after 20 min treatment with Dyn A (1 µM). n = 10 cells. −Lyso/−Nb39 = 23.4 ± 8.1%; −Lyso/+Nb39 = 38.4% ± 17.4%; +Lyso/−Nb39 = 23.7 ± 11.4%; +Lyso/+Nb39 = 14.6 ± 5.1%. (J) cAMP levels after initial Dynorphin treatment (1 µM) for 5 min, washout for 25 min, or a Dynorphin rechallenge (1 µM) at end of the washout, show comparable initial cAMP inhibition by both Dyn A and Dyn B, but persistent signaling by Dyn A after agonist washout.

The online version of this article includes the following figure supplement(s) for figure 3:

**Figure supplement 1.** Differential receptor sorting between Dyn A and Dyn B persists even after agonists are washed out from the surface.

endosomes), EEA1 and Rab5 (early endosomes), Rab11 (recycling endosomes), Rab7 (late endosomes), and Lamp1 (lysosomes), as markers for the biochemically distinct compartments along the early, recycling, and late endosomal pathway. Using automated object-picking, we then quantitated the fraction of KOR that colocalized with each endosomal marker under these conditions. KOR localized mainly to compartments marked by Rab7 and Lamp1 when activated by Dyn A, but to compartments marked by EEA1, Rab5, and Rab11 when activated by Dyn B (*Figure 3B and C*). These results show that KOR is concentrated in different endosomal compartments based on the Dynorphin that activates it.

The agonist-selective localization of KOR to specific endosomes raised the exciting possibility that different Dynorphins could generate distinct subcellular spatial patterns of KOR signaling. To test this possibility, we combined conformation-selective biosensors and high-resolution imaging of FLAG-KOR to ask whether KOR was active in endosomes. A nanobody (Nb39) that specifically recognizes the active conformation of KOR (*Che et al., 2018*), when co-expressed with FLAG-KOR in PC12 cells, localized efficiently to endosomes that also contained Dyn A-activated KOR. In contrast, Nb39 localized less to endosomes containing Dyn B-activated KOR (*Figure 3D and E*). When the fraction of total number of KOR endosomes/cell that colocalized with Nb39 was quantitated by analyzing 3D stacks, endosomes containing Dyn A-activated KOR recruited Nb39 at a significantly higher level (*Figure 3F and G*), suggesting that KOR was in an active conformation in the endosomes specifically after activation by Dyn A. Nb39 recruitment to endosomes required KOR endocytosis, as recruitment was abolished when cells were treated with agonist in the presence of 40 µM Dyngo4A, an endocytosis inhibitor (*Figure 3—figure supplement 1A and B*). To directly examine whether Dyn A-activated KOR in lysosomes was in an active conformation, we used three-color live cell imaging of FLAG-KOR, Nb39, and LysoTracker. In cells treated with 1 µM Dyn A, a subset of KOR endosomes colocalized with both Nb39 and Lysotracker. In cells treated with 1 µM Dyn B, however, virtually no KOR endosomes colocalized with both Nb39 and Lysotracker (*Figure 3H*). When the colocalization was quantitated in Dyn A-treated cells, ~15% of all KOR endosomes colocalized with both markers, suggesting that a subset of Dyn A-activated KOR in the lysosome was in the active conformation (*Figure 3I*). In contrast, in cells treated with 1 µM Dyn B, virtually no KOR endosomes colocalized with both Nb39 and Lysotracker (*Figure 3H*).

To test whether the subset of Dyn A-activated KOR in the active conformation in late endosomes and lysosomes was capable of signaling, we measured cAMP inhibition under conditions where the agonist was washed out to avoid continued signaling from the surface. Twenty-five minutes after agonist washout, Dyn A-activated KOR was still localized predominantly to Rab7-labeled late endosomal compartments, while Dyn B-activated KOR was localized predominantly to Rab11-labeled recycling endosomes (*Figure 3—figure supplement 1C–H*). This distribution was comparable to that observed in the continued presence of agonist, and at this time point, there was little to no KOR degradation (*Figure 2I and J*). Strikingly, Dyn A, but not Dyn B, caused sustained decrease in cAMP levels under conditions where the majority of KOR was in late endosomes and lysosomes (*Figure 3J*). Together, our results suggest that Dyn A, but not Dyn B, specifically coordinates activation and cAMP inhibition by KOR in late endosomes and lysosomes.

Importantly, this Dynorphin-selective coordination of KOR recycling and endosomal activation was conserved in striatal neurons. To directly measure KOR recycling, E18 rat primary embryonic striatal neurons were transfected with SpH-KOR, and individual recycling events were imaged using TIRFM. The number of individual exocytic events, when quantified per minute and normalized to cell area, was significantly lower in neurons treated for 30 min with 1 µM Dyn A compared to Dyn B (*Figure 4A*). This suggested that Dyn B, but not Dyn A, preferentially sorted KOR to recycling

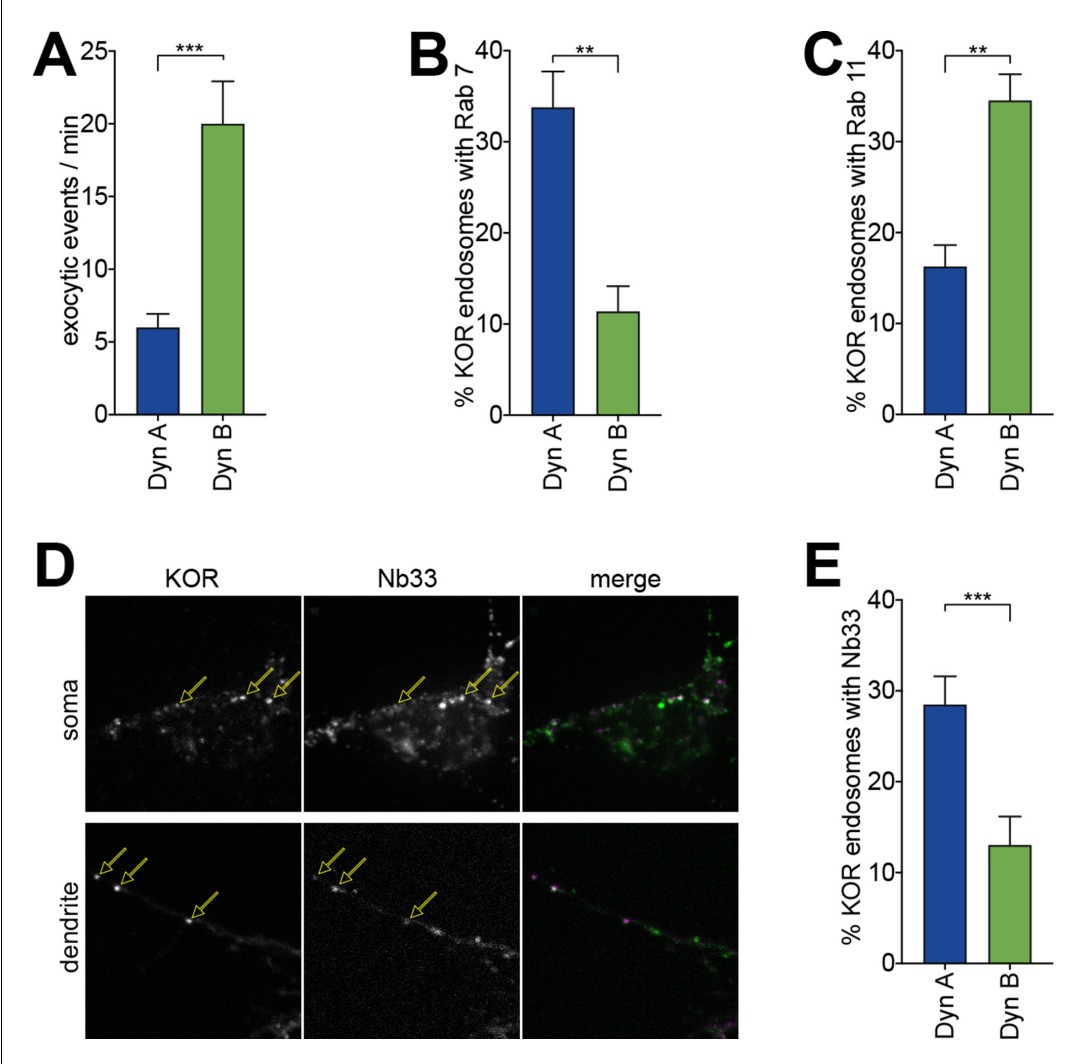

**Figure 4.** Dyn A-specific late endosomal localization and signaling is conserved in striatal neurons. (**A**) The number of discrete exocytic events quantitated in rat medium spiny neuron (MSN) expressing SpH-KOR shows increased recycling for 1 µM Dyn B compared to Dyn A (***p<0.001, n = 8 cells). (**B**) Quantification of the percentage of KOR endosomes colocalized with Rab7 in MSN expressing SpH-KOR treated with 1 µM Dyn A or Dyn B for 30 min (**p<0.01, n = 5 cells for both). (**C**) Quantification of the percentage of KOR endosomes colocalized with Rab11 in MSN expressing SpH-KOR treated with 1 µM Dyn A or Dyn B for 30 min (**p<0.01, n = 5 and 9 cells for Dyn A and Dyn B, respectively). (**D**) Colocalization of FLAG-KOR and Nb33-GFP in the soma and in dendrites of MSNs treated with 1 µM Dyn A for 30 min, seen by confocal microscopy. Yellow arrows show KOR endosomes that recruit Nb33. (**E**) Quantitation of the percentage of KOR endosomes/cell with a noticeable increase in Nb33 fluorescence above background shows that a higher fraction of KOR endosomes recruited Nb33 in Dyn A-treated cells (1 µM for 30 min; ***p=0 < 0.001, n = 10 cells). All p-values were from non-parametric Mann–Whitney tests.

The online version of this article includes the following figure supplement(s) for figure 4:

**Figure supplement 1.** mTOR signaling does not show significant differences between Dyn A and Dyn B.

endosomes in neurons. We directly tested this by detecting the steady-state localization of KOR in endosomes after Dyn A or Dyn B treatment. KOR colocalized predominantly with Rab7 when activated by Dyn A, and with Rab11 when activated by Dyn B (*Figure 4B and C*). To test whether this differential localization correlated with differential location-based activation of KOR in endosomes, we expressed Nb33, a distinct nanobody that recognizes the active conformation of opioid receptors (*Manglik et al., 2017*), fused to GFP in neurons. Endosomes containing Dyn A-activated KOR recruited Nb33, while endosomes containing Dyn B-activated KOR recruited Nb33 to a noticeably lesser extent. This recruitment was readily apparent in dendritic projections, where endosomes were distinctly visible (*Figure 4D*). The percentage of KOR endosomes that recruited Nb33 was

significantly higher for Dyn A-activated KOR than for Dyn B (*Figure 4E*), showing that dynorphin-selective spatial activation of KOR was conserved in neurons.

## Discussion

Together, our results reveal an unanticipated difference between physiologically important endogenous opioid peptides in encoding the subcellular spatial patterns of KOR signaling. Peptidases localized to endosomes, like endothelin-converting enzymes (ECEs) may provide another level of regulation for agonist-dependent KOR trafficking (*Padilla et al., 2007*; *Roosterman et al., 2007*; *Gupta et al., 2015*). However, our results do not suggest that ECE peptide-sensitivity is the only or primary factor that determines agonist-dependent KOR localization (*Figure 2—figure supplement 1*). Ubiquitination of KOR may also regulate its ability to traffic to and signal from late endosomal and lysosomal compartments (*Li et al., 2008*; *Henry et al., 2011*; *Dores and Trejo, 2019*).

The exact mechanism by which KOR is localized to different compartments is not clear. Because the post-endocytic sorting of GPCRs is usually mediated by specific interactions of the unstructured cytoplasmic tail of the receptors with trafficking proteins, KOR sorting probably involves interactions with PDZ-interacting protein NHERF1/EBP50 (*Liu-Chen, 2004*). In this context, different Dynorphin peptides could lock KOR into conformations that selectively allow or inhibit interactions with trafficking and signaling proteins, essentially defining the receptor interactome in an agonist-specific manner. This conformational lock could require the presence of the ligand that is co-internalized with the receptor, although an exciting possibility is that the ligands provide a 'conformational memory' to KOR that is sustained through the endocytic trafficking pathway. At present it is not clear what could provide such a conformational memory. It is possible that different Dynorphins could cause agonist-specific post-translational modifications, which is a general emerging theme for opioid receptors (*Chiu et al., 2017*; *Mann et al., 2019*). In any case, the differential subcellular localization and trafficking of KOR by two physiologically important ligands that we show here are important to underscore the physiological relevance of receptor sorting, which has been studied largely using receptor mutants or by depleting key components of the trafficking machinery (*Bowman et al., 2016*; *Weinberg et al., 2019*; *Zhao et al., 2013*; *Sposini et al., 2017*).

One striking aspect of our results is that Dyn A-bound KOR activates Gi in the late endosomal pathway on its way to being degraded. This is surprising because early endosomes are the main compartments that support endosomal signaling for most other canonical GPCRs. For example, other Gi-coupled receptors such as the mu opioid receptor can exist in an active conformation in earlier endosomal compartments (*Stoeber et al., 2018*). Internalization is required for sustained inhibition of cAMP for many receptors, such as for the class B S1P receptor (*Willinger et al., 2014*). However, Gi is likely present on late endosomal and lysosomal compartments, as cannabinoid receptors trafficking from the Golgi can activate Gi on lysosomes (*Rozenfeld and Devi, 2008*). Further, the binding of peptides to opioid receptors does not change dramatically at lower pH (*Gupta et al., 2015*). Therefore, it is possible that Gi could be activated at multiple endosomal compartments based on the specific opioid receptor and ligand, leading to distinct early and late phases of endosomal signaling.

Post-translational modifications such as phosphorylation or ubiquitination could provide regulatory handles for this late phase of signaling. For example, three lysine residues on the C-terminus of KOR are required for normal levels of degradation of KOR, but not for internalization from the surface (*Li et al., 2008*). Ubiquitination, however, plays complex roles in receptor trafficking and signaling at the endosome, controlling transport of receptors to lysosomes, entry of receptors into intralumenal vesicles, and recruitment of signaling scaffolds that could initiate non-canonical signaling pathways (*Patwardhan et al., 2021*). In this context, Dyn A-activated KOR could also activate alternate pathways, such as mTOR signaling, on late endosomes and lysosomes. Interestingly, mTOR signaling is involved in potentially deleterious effects of KOR, leading to efforts to generate agonists bypassing this signaling pathway (*Che and Roth, 2018*; *Liu et al., 2018*; *Liu et al., 2019*). However, under the conditions we tested, we did not see a significant increase in p70S6 phosphorylation, as a readout for mTOR activation, upon activation of KOR with either Dyn A or Dyn B. This could potentially be due to the high baseline of p70S6 phosphorylation in the PC12 cells (*Figure 4—figure supplement 1*). However, it is possible that KOR could activate mTOR signaling from late endosomes or lysosomes in a subset of neurons yet to be identified, that could mediate aversive effects of KOR.

Physiological systems could leverage receptor sorting to fine-tune both spatial and temporal aspects of GPCR signaling. KOR is activated by many opioid peptides that are generated from multiple precursor peptidesproteins, some of which bias signaling from the cell surface to different outputs (*Gomes et al., 2020*). Our results suggest that, even for peptides where there are no obvious differences in surface signaling, there are differential effects in endocytic sorting and signaling from endosomes. Receptor sorting in cells is a dynamic and incomplete process. The fractions of receptors we see at steady state likely represent an equilibrium of many rounds of rapid iterative sorting as the endosome matures, where only a small fraction is recycled back to the surface in the case of Dyn A, while a large fraction is recycled in the case of Dyn B. Because Dyn A drives little KOR to recycle and promotes endosomal KOR activation, the net effect would be to cause a sustained cAMP inhibition from endosomes after a single exposure. Because Dyn B drives KOR recycling and induces endosomal signaling only to small amounts, the net effect would be short-lived cAMP inhibition primarily from the surface. On the other hand, the rapid recycling and resensitization caused by Dyn B would sensitize cells to repeated pulses of ligand release, unlike with Dyn A. This difference in steady-state localization, however, is enough to cause a difference in endosomal receptor activation and cAMP signaling, suggesting that small differences in steady-state localization can cause relevant changes in signaling.

Whether these different Dynorphins are always co-released in the nervous system, or whether different brain regions selectively release specific Dynorphins, is still unclear. Dyn A and Dyn B are generated from prodynorphin likely in the late stages of dense core vesicle maturation and could be predominantly co-released, but there could be mechanisms that actively segregate or selectively release individual Dynorphins. In the case of co-released peptides, it is possible that one or more of the peptides could be dominant in dictating the conformational states in which receptors spend most of their time, in which case the signaling and trafficking fates would be determined primarily by these dominant peptides. In any case, our results that highly related opioid peptides regulate spatial encoding of KOR suggest an unanticipated layer of granularity to the anatomical and functional maps of the brain.

## Numerical data file

An Excel file reporting the numerical data (means, standard deviations, standard errors, p-values, and n) for the graphs in the figures as noted is provided as *Source data 1*.

# Materials and methods

## Reagents, constructs, and cells

Dynorphin A17 (Dyn A), Dynorphin B13 (Dyn B), Dynorphin A (Dyn A8), and α-neoendorphin (α-neo) were purchased from Tocris Bioscience and/or Phoenix Pharmaceuticals. Naltrexone, protease inhibitor cocktail (Cat. No. P2714), anti-Flag M2 antibody (Cat. No. F3165) were purchased from Sigma Aldrich (St. Louis, MO). Anti-APPL1, -EEA1, -Rab5, -Rab11, -Rab7, -Lamp1 rabbit monoclonal antibodies were purchased from Cell Signaling Technology. Anti-GFP rabbit polyclonal antibodies (Cat. No. A10260) were from Thermo Fisher Scientific. Nb39 and Nb33 constructs were provided by Dr. Bryan Roth (UNC Chapel Hill) and Dr. Mark von Zastrow (UCSF) respectively. Cell lines used were validated, and cells were purchased from ATCC. Cells in the lab are routinely tested for mycoplasma contamination. Stable non-clonal PC12 cells expressing KOR N-terminally tagged with supereccliptic phluorin (SpH) (SpH-KOR cells) were selected in puromycin (Gibco) and grown in F12K media supplemented with 10% horse serum and 5% fetal bovine serum (Gibco) in collagen coated flasks. PC12 cells were also transiently transfected with KOR fused to FLAG on its N-terminus using Lipofectamine 2000 as per manufacturer's protocol (ThermoFisher). Transfected cells were imaged 2–3 days after transfection. CHO cells stably expressing Flag epitope tagged KOR generated as described previously (*Gupta et al., 2016*) were grown in F12 media supplemented with 10% fetal bovine serum and 1% penicillin–streptomycin. E18 rat striatal neurons were obtained from BraintBits LLC and cultured on poly-d-lysine (Sigma) coated coverslips for 1 week in Neurobasal media (Gibco) supplemented with B27 (Gibco), 1% Glutamax (Gibco), and 1% penicillin–streptomycin (Gibco) before transfection with SpH-KOR or Flag-KOR and Nb33-GFP using Lipofectamine 2000 as per manufacturer's protocol (ThermoFisher). Antibodies used are listed below.

| Antibody | Source | Identifier |
|---|---|---|
| Rabbit anti-APPL1 | Cell Signaling Technology | D83H4 XP, #3858S, Lot 1 |
| Rabbit anti-EEA1 | Cell Signaling Technology | C45B10, #3288, Lot 2 |
| Rabbit anti-Rab5 | Cell Signaling Technology | C8B1, #3547, Lot 7 |
| Rabbit anti-Rab11 | Cell Signaling Technology | D4F5 XP, #5589, Lot 4 |
| Rabbit anti-Rab7 | Cell Signaling Technology | D95F2 XP, #9367, Lot 1 |
| Rabbit anti-Lamp1 | Cell Signaling Technology | D2D11 XP, #9091, Lot 6 |
| Goat anti-Rabbit IgG Secondary Antibody, Alexa Fluor 647 | Invitrogen | A21244, Lot 792514 |
| Mouse anti-FLAG M1 (647 conjugate) | Invitrogen | A20173A, Lot 2136857 |
| Mouse anti-FLAG M2 | Sigma | F3165 |
| Rabbit anti-GAPDH | Cell Signaling Technology | 14C10, #2118S, Lot 14 |
| Rabbit p70 S6 Kinase | Cell Signaling Technology | 9202S, Lot 20 |
| Rabbit phospho-p70 S6 Kinase (Thr389) (108D2) | Cell Signaling Technology | 9234S, Lot 12 |
| Chicken anti-GFP | Abcam | Ab13970, Lot GR3190550-10 |
| Goat anti-Rabbit IgG (H + L)-HRP conjugate | BioRad | 170–6515, L005679A |

## Displacement binding assays

Displacement binding assays were carried out using membranes from PC12 cells stably expressing SpH-KOR (SpH-KOR cells) (100 μg) and CHO-KOR (15 μg) cells. Membranes were prepared as described previously (*Gomes et al., 2003*). Displacement binding assays were carried out as described previously (*Gomes et al., 2004*; *Gomes et al., 2011*) by incubating membranes with [$^3$H] diprenorphine (3 nM) without or with different concentrations ($10^{-12}$ to $10^{-5}$ M) of Dyn A8, Dyn A17, Dyn B13, or a-neo-endorphin in 50 mM Tris-Cl buffer pH 7.4 containing 100 mM NaCl, 10 mM MgCl$_2$, 0.2 mM EGTA, and protease inhibitor cocktail (Sigma-Aldrich; cat No. P2714) for 1 hr at 37°C. Non-specific binding was determined in the presence of 10 μM cold diprenorphine. Specific Bound Counts obtained in the absence of peptides was taken as 100%. Data presented are mean ± SE of three independent experiments in triplicate.

## Live cell imaging

Cells were plated onto poly-D-lysine (Sigma) coated 25 mm coverslips. Cells were imaged 2 days later in Leibovitz L15 imaging medium (Gibco) and 1% fetal bovine serum at 37°C in a CO-controlled imaging chamber, using a Nikon Eclipse Ti automated inverted microscope with a 60× or a 100 × 1.49 N.A. TIRF objective or a 20 × 0.75 N.A. objective. Images were acquired with an iXon +897 electron-multiplying charge-coupled device camera with a solid state laser of 488 nm or 647 nm as a light source. Images were analyzed using FIJI (*Schindelin et al., 2012*).

## Quantification of individual recycling events

PC12 cells stably expressing SpH-KOR (SpH-KOR cells) were treated with KOR agonists: Dyn A17, Dyn B13, Dyn A8, or α-neo endorphin (1 μM) for 5 min to induce receptor clustering and internalization at 37°C. Receptor clustering was visualized using TIRF microscopy. Images were acquired every 3 s for a total of 5 min. Following internalization, a recycling movie was recorded at 10 Hz for 1 min in TIRF. The number of exocytic recycling events were manually scored in FIJI (Fiji Is Just Image J) to determine the recycling rate for each agonist. Recycling events were counted throughout the 1 min movie and the total number of events were normalized by the cell area to determine a recycling rate. Recycling events were also recorded using the same method in primary striatal rat medium spiny neurons transfected with the SpH-KOR plasmid. Recycling movies were taken 30 min after agonist addition in neurons. Statistical significance was determined using a one-way ANOVA.

## Ensemble recycling assay

Receptor surface levels were measured in PC12 cells stably expressing SpH-KOR (SpH-KOR cells) by using confocal microscopy on a 20× objective and 488 nm laser. Images were collected in 30 s intervals across 20 different cell fields. After 2 min of baseline an agonist (1 µM Dyn A17 or Dyn B13) was added to imaging media. Following agonist addition, images were collected for 15 min. After 15 min, agonist was removed, and the imaging media was replaced with fresh media containing antagonist (naltrexone; 10 µm). Images were then collected for another 15 min. Fluorescence intensities were corrected by a background threshold and normalized by the average fluorescence of the baseline frames before agonist treatment. Surface fluorescence analysis was conducted using an ImageJ Macro automated script (National Institutes of Health) (*Weinberg et al., 2019*). Fluorescence recovery/loss ratios after washout were quantified by normalizing the fluorescence values after washout to the total fluorescence lost before washout. Cell fields that did not respond to Dynorphin treatment were excluded from analysis. Statistical significance was determined by using Student's paired t-test comparing the endpoints between agonist treatment.

## ELISA internalization assays

CHO cells expressing Flag-epitope tagged KOR (CHO-KOR cells) or PC12 cells stably expressing SpH-KOR (SpH-KOR cells) were seeded in complete growth media into 24-well plates ($2 \times 10^5$ cells per well). Next day, cells were rinsed with PBS followed by labeling with mouse anti-Flag antibodies for CHO-KOR cells or chicken anti-GFP antibodies for SpH-KOR cells (1:1000 in PBS containing 1% BSA) for 1 hr at 4°C, followed by treatment with 0–10 µM of Dyn A or Dyn B in growth media containing protease inhibitor cocktail (Sigma-Aldrich; Cat. No. P2714) for 60 min at 37°C. Cells were briefly fixed (3 min) with 4% paraformaldehyde followed by three washes (5 min each) with PBS, and incubation with anti-mouse or anti-chicken antibody coupled with horse-radish peroxidase (1:1000 in PBS containing 1% BSA) for 90 min at 37°C. Cells were washed three times with 1% BSA in PBS (5 min each wash), and color was developed by the addition of the substrate *o*-phenylenediamine (5 mg/10 ml in 0.15 M citrate buffer [pH 5] containing 15 µl of $H_2O_2$). Absorbance at 490 nm was measured with a Bio-Rad ELISA reader. Values obtained with secondary antibody in the absence of primary antibody were taken as non-specific and subtracted from all points. The percentage of internalized receptors was calculated by taking total cell surface receptors before agonist treatment for each individual experiment as 100% and subtracting percent surface receptors following agonist treatment. Data presented are mean ± SE of three independent experiments in triplicate.

## ELISA recycling assays

CHO cells expressing Flag-epitope tagged human KOR (CHO-KOR cells) or PC12 cells stably expressing SpH-KOR (SpH-KOR cells) were seeded in complete growth media into 24-well plates ($2 \times 10^5$ cells per well). Next day, cells were rinsed with PBS followed by labeling with mouse anti-Flag antibodies for CHO-KOR cells or chicken anti-GFP antibodies for SpH-KOR cells (1:1000 in PBS containing 1% BSA) for 1 hr at 4°C, followed by treatment with 100 nM Dyn A, Dyn B, or BAM-22 in growth media containing protease inhibitor cocktail (Sigma-Aldrich; Cat. No. P2714) for 30 min to elicit receptor internalization. The cells were washed to remove the agonist and incubated with medium without or with the ECE2 inhibitor (S136492, 20 µM) for 0–120 min to allow for receptor recycling. At the end of the incubation period, cells were chilled to 4°C and then fixed briefly (3 min) with 4% paraformaldehyde followed by three washes (5 min each) with PBS and incubation with anti-mouse or anti-chicken antibody coupled with horse-radish peroxidase (1:1000 in PBS containing 1% BSA) for 90 min at 37°C. Cells were washed three times with 1% BSA in PBS (5 min each wash), and color was developed by the addition of the substrate *o*-phenylenediamine (5 mg/10 ml in 0.15 M citrate buffer [pH 5] containing 15 µl of $H_2O_2$). Absorbance at 490 nm was measured with a Bio-Rad ELISA reader. Values obtained with secondary antibody in the absence of primary antibody were taken as non-specific and subtracted from all points. % recycled receptors were calculated by subtracting receptors at t = 0 (30 min internalization) from each recycling time point; this represents 0% recycled receptors. Data presented are mean ± SEM of three independent experiments in triplicate.

## Immunofluorescence of endosomal markers

PC12 cells stably expressing SpH-KOR were plated on poly-d-lysine (Sigma Aldrich) coverslips and grown for 24–48 hr at 37°C. Cells were then incubated with different agonists (Dyn A, Dyn B, Dyn A8, or α-neo) for 20 min at 37°C. Cells were then fixed with 4% paraformaldehyde (PFA), pH 7.4, for 20 min. Cells were then rinsed with complete PBS twice and then blocked in PBS containing calcium, magnesium, with 5% FBS, 5% 1M glycine, and 0.75% Triton X-100. SpH-KOR cells were then incubated with an antibody for one of the endosomal markers for 1 hr. Cells were washed three times with PBS containing calcium and magnesium and then labeled with Alexa 647 goat anti-rabbit secondary antibody (1:1000) in a blocking buffer for 1 hr. Confocal imaging of cells was performed using spinning disk confocal microscope (Andor) and 100× objective. Representative images were taken across 10–20 fields for each agonist treatment and endosomal marker. Three biological replicates were performed in each condition.

## Endosomal KOR colocalization in live cells with nanobodies and lysotracker

PC12 cells were transiently transfected with FLAG-KOR and Nb39-YFP or (Nb33-GFP). Cells were labeled with M1-647 for 10 min prior to imaging 3 days after transfection. Images were taken before and after cells were treated with 1 μM Dyn A or Dyn B for 20 min. Confocal imaging of cells was performed using spinning disk confocal microscope (Andor) and 100× objective. Representative images were taken across 10–20 fields for each agonist treatment. In the experiments with Dyngo4A, cells were pretreated with 40 μM Dyngo4a for 30 min prior to imaging. In the Lysotracker experiments, cells were labeled with 25 nM Lysotracker-561 for 5 min prior to imaging.

## Endosomal colocalization quantification

The percent colocalization of the endosomal marker with the total number of receptor positive endosomes was determined using an ImageJ Macro: Object.picker (*Weinberg, 2020*; doi.10.5281/zenodo.3811031) to identify the total number of endosomes containing receptor in one channel and determine the colocalization with an endosomal marker in another channel. The Image J macro: 3D Object Counter was used as another method of quantification for colocalization. Integrated density values for each object detected in both the receptor and endosome marker channels were used to determine a ratio of endosomal colocalization by dividing the endosomal marker signal by the receptor signal.

## Immunoblotting

PC12 cells stably expressing SpH-KOR were grown in a PDL coated 12-well plate for 2 days at 37°C. Cells were treated with cycloheximide (3 μg/ml) for 2 hr before agonist incubation. Cells were treated with Dyn A17 or Dyn B13 for 30 min or 2 hr. A non-agonist treated well of PC12 cells stably expressing SpH-KOR and a well of PC12 cells not expressing SpH-KOR were used as controls. Following agonist treatments, cells were placed on ice and rinsed twice with PBS containing calcium and magnesium. Cells were directly lysed in the plate using 2× RSB (Bio-Rad, Hercules, CA). Lysates were placed on ice for 30 min and then sonicated in 5 s pulses. Following sonication, lysates were incubated at 37°C for 1 hr. Lysates were run on 10% stain-free gels (BioRad), which were then transferred to nitrocellulose membrane overnight. Membranes were blocked in 5% milk and then probed with anti-GFP Chicken pAB (Abcam) to detect total receptor levels in each condition. Blots were developed using the iBright imager for chemiluminescence signal and quantified using FIJI software. Receptor signal for each condition was normalized to the no treatment control. Five biological replicates were performed. Statistical analysis was performed using two-way ANOVA across time and drug treatment. To test for mTOR activation, PC12 cells stably expressing SpH-KOR were grown in a PDL coated 12-well plate for 2 days at 37°C. Cells were starved overnight in serum-free media and then treated with 1 μM Dyn A or Dyn B for 5 min or 20 min. Cells were placed on ice and rinsed twice with PBS containing calcium and magnesium. Cells were directly lysed in the plate using 2× RSB (Bio-Rad, Hercules, CA). Lysates were placed on ice for 5 min and then placed at 95°C for 5 min. Lysates were run on 10% stain-free gels (BioRad), which were then transferred to nitrocellulose membrane overnight. Membranes were blocked in 5% BSA and then probed with phospho-p70 S6K (CST) to detect phosphorylated S6K levels in each condition. Blots were developed using the iBright

imager for chemiluminescence signal and quantified using FIJI software. Membrane was stripped and probed with total p70 S6K (CST) to determine total levels of S6K present in the samples. The phospho-p70 S6K signal was normalized to the total p70 S6K signal for each condition. All samples were then normalized to the no treatment control to determine the fold change over baseline for each condition. Five biological replicates were performed. Statistical analysis was performed using two-way ANOVA across time and drug treatment.

### cAMP assays

PC12 cells stably expressing SpH-KOR (SpH-KOR cells) or CHO cells stably expressing Flag epitope tagged human KOR (CHO-KOR cells) cells (10,000/well) were treated with Dyn A, Dyn B, Dyn A8, or α-neo (1 µM) for 30 min at 37˚C in HBSS assay buffer containing 10 mM HEPES, 20 µM forskolin, and protease inhibitor cocktail (Sigma-Aldrich; Cat. No. P2714) and cAMP levels were quantified using the HitHunter cAMP detection kit from DiscoveRx according to the manufacturer's protocol. In a separate set of experiments dose–response curves were carried out with Dyn A or Dyn B (0–10 µM). In another set of experiments cells were treated Dyn A or Dyn B (1 µM) for 5 min, after which peptides were washed out and cells were incubated in assay buffer for 25 min. Cells were then given a second 5 min treatment with Dyn A or Dyn B (1 µM) and cAMP levels measured. Values obtained in the absence of peptide were taken as 100%. Data presented are mean ± SEM of three independent experiments in triplicate.

## Acknowledgements

We thank Dr. Daniel Shiwarski, Marlena Darr, and Caroline Hernandez-Casner for essential initial technical assistance with the project. We thank Drs. Bryan Roth, Tao Che, Daniel Wacker, Mark von Zastrow, and Miriam Stoeber for generously providing key reagents. We thank Drs. Robert Fuller, Carole Parent, Alan Smrcka, and Lloyd Fricker for expert discussions. JMK was supported by NIH T-32-GM007315, LAD by NIH NS026880 and DA008863, and MAP by NIH GM117425 and by NSF 1935926.

## Additional information

### Funding

| Funder | Grant reference number | Author |
| --- | --- | --- |
| National Institute of General Medical Sciences | T32GM007315 | Jennifer M Kunselman |
| National Institute of General Medical Sciences | GM117425 | Manojkumar A Puthenveedu |
| National Science Foundation | 1935926 | Manojkumar A Puthenveedu |
| National Institute of Neurological Disorders and Stroke | NS026880 | Lakshmi A Devi |
| National Institute on Drug Abuse | DA008863 | Lakshmi A Devi |

The funders had no role in study design, data collection and interpretation, or the decision to submit the work for publication.

### Author contributions

Jennifer M Kunselman, Conceptualization, Formal analysis, Validation, Investigation, Visualization, Methodology, Writing - original draft, Writing - review and editing; Achla Gupta, Ivone Gomes, Conceptualization, Formal analysis, Validation, Investigation, Visualization, Methodology, Writing - review and editing; Lakshmi A Devi, Conceptualization, Resources, Supervision, Funding acquisition, Investigation, Methodology, Project administration, Writing - review and editing; Manojkumar A Puthenveedu, Conceptualization, Resources, Formal analysis, Supervision, Funding acquisition, Visualization, Methodology, Project administration, Writing - review and editing

## Author ORCIDs

Jennifer M Kunselman (ID) https://orcid.org/0000-0002-9772-4149
Manojkumar A Puthenveedu (ID) https://orcid.org/0000-0002-3177-4231

## Decision letter and Author response

Decision letter https://doi.org/10.7554/eLife.60270.sa1
Author response https://doi.org/10.7554/eLife.60270.sa2

## Additional files

### Supplementary files

- Source data 1. Numerical data.

- Transparent reporting form

### Data availability

Data generated and analyzed in this study are included in the manuscript. The study did not generate new sequencing or structural data.

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
