## [Decision Letter]

**Acceptance summary:**

Why there are over 20 different endogenous opioid peptides but only four receptors, has been a question that has been unanswered for decades. Here the authors show that two highly related endogenous opioids, which initially activate kappa opioid receptor signaling to similar levels, diverge subsequently in trafficking and endosomal signaling. Demonstrating that different endogenous opioids can differentially regulate localization and trafficking of (and signaling by) the same receptor is a key advance in the opioid field.

**Decision letter after peer review:**

[Editors’ note: the authors submitted for reconsideration following the decision after peer review. What follows is the decision letter after the first round of review.]

Thank you for submitting your work entitled "Compartment-specific opioid receptor signaling is selectively modulated by Dynorphin subtypes" for consideration by *eLife*. Your article has been reviewed by 3 peer reviewers, and the evaluation has been overseen by a Reviewing Editor and a Senior Editor. The following individual involved in review of your submission has agreed to reveal their identity: Christopher Evans (Reviewer #3).

Our decision has been reached after consultation between the reviewers. Based on these discussions and the individual reviews below, we regret to inform you that your work will not be considered further for publication in *eLife*.

This is an interesting and creative paper implicating a differential mechanism of intracellular trafficking and subsequent signaling that is triggered by different dynorphins binding to the kappa opioid receptor. In principle, if the authors could explain the molecular basis for this phenomenon, the story would be of tremendous impact in the fields of opioid receptor signaling and trafficking. Unfortunately, the reviewers noted a number of concerns that would require significant further work and clarification to support the authors' conclusions at this stage. We hope you will find the comments constructive as you plan your next steps.

*Reviewer #1*

In this manuscript the authors have assessed the different endocytic routes of KOR when activated by DynA or DynB. These are nicely conducted experiments that show interesting results, however the authors completely obviate the connection with their own work that highlights the different degradation mechanisms of these two peptides. As it stands it does not add to the field, and lacks a mechanistic explanation that could be explored given the authors expertise in these systems.

1. The major conclusion of the study is that after endocytosis, DynA preferentially sorts KOR into the degradative pathway, while DynB sorts KOR into the recycling pathway and that this has consequences in the duration of the active state of the receptor and its ability to signal. It is surprising that the authors do not investigate the connection between these results and previously published work that shows differences in the degradation of DynB vs DynA within endosomes. Indeed, the authors have previously shown that: (i) ECE2 hydrolizes DynB and not DynA (Mzhavia et al., JBC 2003), (ii) overexpression of ECE2 increases the rate of mu-opioid receptor recycling upon DynB stimulation (Gupta et al., BJP 2015) and (iii) inhibition of ECE2 decreases mu-opioid receptor recycling (Gupta et al., BJP 2015). Considering this previous work, it is totally expected that the two ligands show distinct post-endocytic trafficking of KOR.

2. Similarly, the differences in ECE2 sensitivity can also explain the Nb39 results, with KOR activated by the ligand that is not hydrolysable (DynA) being able to remain in the active state (and signal) for longer than when activated with the hydrolozable ligand (DynB).

3. A simple experiment to address this obvious connection is to use an ECE2 inhibitor. One would expect that in the presence of this inhibitor DynB-activated KOR is retained intracellularly and remains active for longer. This is an obvious omission of this work and in my opinion the manuscript should not be published without some experiments addressing this.

4. The authors state "this is the first example of different physiological agonists driving spatial localization and trafficking of a GPCR" in light of the above comment, previous work from Bunnett et al., have shown how peptides with different endocytic enzyme sensitivity can indeed, localize GPCRs (e.g somatostatin receptor) in different compartments and elicit distinct signals (Padilla et al., J Cell Biol 2007; Roosterman et al., PNAS 2007; Zhao et al., JBC 2013 to name a few).

5. Support for endosomal signalling falls a bit short. For example, if indeed KOR signals from endosomes, the authors should use an inhibitor of receptor internalization and assess Nb39 recruitment and KOR signalling. Although this reviewer agrees that, since this experiment would still provide indirect evidence, it may not be necessary (though desirable) for publication.

*Reviewer #2*

This manuscript demonstrates that two highly similar endogenous opioid agonists can give distinct opioid receptor trafficking and signaling fates. There are two key observations that are novel and intriguing: (1) two opioid peptides that are derived from the same precursor can distinctly modulate Kappa Opioid receptor (KOR) trafficking into two distinct pathways; Dynorphin A causes KOR trafficking to the late endosomes/lysosomes pathway whereas Dynorphin B promotes rapid recycling; (2) Dynorphin A activates Gi proteins on the late endosomes/lysosomes which leads to Gi-mediated cAMP inhibition from these compartments.

The idea that GPCRs can activate G proteins at the late endosome/lysosomal compartments is fascinating and novel, however, the data presented here does not fully support their model that Dynorphin A activated Gi proteins on the late endosomes/lysosomes. Without better support for this model, publication in *eLife* would not be warranted. Here are the main questions that should be addressed before publication:

1. There is a mismatch with the timing of receptor colocalization experiment (Figure 3B and C, 20 min Dynorphin A/B treatment) and the cAMP assay (Figure 3H, 5 min treatment). There needs to be direct evidence that KOR is localized on the late endosomes/lysosomes at 5 minutes post agonist stimulation, i.e. at the time that cAMP levels are measured. It is important to demonstrate that the sustained signaling inhibition by DynA comes from the late endosomes/lysosomes as opposed to early endosomes. A colocalization experiment with 5 min DynA stimulation followed by a 25min washout would be necessary to support their model.

2. What percentage of KORs are proteolytically degraded in the late endosomes/lysosomes at 20 min DynA stimulation?

3. Given that KOR trafficking to the late endosomes and lysosomes is mediate by ubiquitination (as shown here PMID: 18212250), does mutation of these ubiquitination sites (3 lysine residues on KOR C-terminus) block its trafficking and the sustained signaling from the late endosomes/lysosomes?

4. Is there any evidence for Gi protein localization on the late endosome/lysosomes?

5. Additional functional readouts would also be helpful to support their model of Gi-mediated inhibition of cAMP response from late endosomes/lysosomes and not the plasma membrane or early endosomes. Perhaps mTOR activation (as authors have suggested in their discussion) could be used as a read out to show differences between DynA and B-mediated signaling?*Reviewer #3*

This is an interesting idea and creative paper implicating a differential mechanism of intracellular trafficking and subsequently signaling that is triggered by different dynorphins binding to the kappa opioid receptor. However, there are some questions for the authors:

1. My reading is that some dynorphins are extremely rapidly degraded in serum and with these experiments performed in 15% Horse/FCS there is concern that some of the differential results could be explained by differential degradation. One hypothesis could be a differential frequency of receptor activation over time of a fast recycling receptor population. Can the authors convince me that this difference in trafficking and subsequent signaling is an intrinsic property of the peptide and not an exhaustion of peptide (would be DynB) over the 30min assay?

2. In Figure 2D, 2G and 2J at what time after addition peptides was this data obtained?

3. In Figure 2F the divergence of internalized receptor only occurs from time 20-30 mins which was difficult for me to understand since DynA should be resulting in lost surface receptor number. What confuses me is that in Fig2H the initial recycling induced by DynA17 is fast and slows down so I am wondering if a second hit is needed which feeds into my concern about peptide degradation in the media. Since released peptide would be pulsatile maybe in vivo DynA17 could act like DynB?

4. The assays seem to be done with a single concentration of peptide – 1µM. Do the authors have data to show that at lower (or higher) concentrations than 1µM result in the same trafficking patterns, albeit to a lesser or greater extent. Also, for the cAMP inhibition what concentration gives max inhibition? For a binding affinity of 0.01nM in the cells and with high expression, the 1micromolar concentration seem high.

5. In Figure 2H 100% of receptors appear to be recycled after DynB however 25% of kappa colocalize in Rab7 in 3C so do these Rb 7 co-localized receptors recycle?

6. Could some of the signaling differences be explained by continued activation of receptor as a consequence of peptide processing in the endocytosed vesical as opposed to different vesicles? I guess the continued signaling could also direct subsequent trafficking and this could be tested with a membrane permeable antagonist.

7. The impact statement" Co-released dynorphins, which signal similarly from the cell surface, can differentially localize GPCRs to specific subcellular compartments, and cause divergent receptor fates and distinct spatiotemporal patterns of signaling" could be misconstrued. If one of the pathways is dominant and blocks the other, then co-release may only have one signaling outcome. Have any dynorphin mix experiments been conducted? What might be anticipated?

8. It looks like details for the ELISA measurements in the methods section was missing. Were the ELISA measurements done with untagged KOR or SpH-KOR? One might worry about the effects of the N-terminal SpH tag on KOR trafficking, and it would be nice if the fluorescence SpH-KOR data were supported by ELISA for untagged KOR. (At least some of the data is immunostaining of FLAG-KOR, which probably introduces only minimal perturbation).

9. Dynorphin A17 is a very sticky peptide and difficult to wash out. Since we don't have a dose response it may require only very doses to have full activation for cAMP inhibition. It would be nice to be able to discount this as a potential for prolonged activation after washout.

---

## [Author Response]

[Editors’ note: The authors appealed the original decision. What follows is the authors’ response to the first round of review.]

This is an interesting and creative paper implicating a differential mechanism of intracellular trafficking and subsequent signaling that is triggered by different dynorphins binding to the kappa opioid receptor. In principle, if the authors could explain the molecular basis for this phenomenon, the story would be of tremendous impact in the fields of opioid receptor signaling and trafficking. Unfortunately, the reviewers noted a number of concerns that would require significant further work and clarification to support the authors' conclusions at this stage. We hope you will find the comments constructive as you plan your next steps.

We are very happy that you and the reviewers found that the study could be of tremendous impact and describe the paper as “interesting and creative”, “novel and intriguing”, “fascinating and novel”, and feel that the study was “nicely conducted”. We appreciate the comments of the reviewers, and we have addressed the comments as described below.

Reviewer #1In this manuscript the authors have assessed the different endocytic routes of KOR when activated by DynA or DynB. These are nicely conducted experiments that show interesting results, however the authors completely obviate the connection with their own work that highlights the different degradation mechanisms of these two peptides. As it stands it does not add to the field, and lacks a mechanistic explanation that could be explored given the authors expertise in these systems.

We thank the reviewer for the positive comments. We are happy that the reviewer felt that the experiments are nicely conducted, and that the results are interesting. However, we respectfully but strongly disagree with the comments that our study does not add to the field.

First, considering the extended and severe opioid epidemic, understanding the many ways in which the opioid peptide/receptor system is modulated is of high priority. Endogenous opioid peptides are highly relevant neuromodulators about which we know even less than opioid drugs. Why there are over 20 different endogenous opioid peptides but only four receptors, has been a question that has been unanswered for decades. We show that two highly related endogenous opioids, which initially activate KOR to similar levels, diverge subsequently in trafficking and endosomal signaling. The importance of opioid receptor trafficking to signaling and tolerance is well accepted (although the mechanisms are debated). Demonstrating that different endogenous opioids can differentially regulate localization and trafficking of (and signaling by) the same receptor is a key advance in the opioid field.

Second, the idea that location based-biased signaling can lead to different consequences for the same agonist is a relatively new idea, and clearly a very important area of continuing research. Even for well-studied systems like the adrenergic receptor system, we know almost nothing about the physiological relevance of differential signaling because most of the data are from synthetic compounds. Demonstrating that endogenous ligands take advantage of location bias to generate distinct signaling consequences is a clear indication that such differential trafficking and signaling is physiologically relevant, which is a clear advance in the GPCR field.

1. The major conclusion of the study is that after endocytosis, DynA preferentially sorts KOR into the degradative pathway, while DynB sorts KOR into the recycling pathway and that this has consequences in the duration of the active state of the receptor and its ability to signal. It is surprising that the authors do not investigate the connection between these results and previously published work that shows differences in the degradation of DynB vs DynA within endosomes. Indeed, the authors have previously shown that: (i) ECE2 hydrolizes DynB and not DynA (Mzhavia et al., JBC 2003), (ii) overexpression of ECE2 increases the rate of mu-opioid receptor recycling upon DynB stimulation (Gupta et al., BJP 2015) and (iii) inhibition of ECE2 decreases mu-opioid receptor recycling (Gupta et al., BJP 2015). Considering this previous work, it is totally expected that the two ligands show distinct post-endocytic trafficking of KOR.2. Similarly, the differences in ECE2 sensitivity can also explain the Nb39 results, with KOR activated by the ligand that is not hydrolysable (DynA) being able to remain in the active state (and signal) for longer than when activated with the hydrolozable ligand (DynB).3. A simple experiment to address this obvious connection is to use an ECE2 inhibitor. One would expect that in the presence of this inhibitor DynB-activated KOR is retained intracellularly and remains active for longer. This is an obvious omission of this work and in my opinion the manuscript should not be published without some experiments addressing this.

First of all, we respectfully, but strongly, disagree with the implication that only unexpected results are worth publishing. Differentiating and establishing predicted outcomes is critical for advancing biology. Demonstrating a logical conclusion is an essential part of this process. This should be viewed as a strength, and not as a weakness. Acknowledging and supporting this idea is especially important in these times where there is a strong effort and an opportunity, led by *eLife*, to make academic publishing open and fair.

In this case, however, the key to the reviewer’s three concerns is the assumption that the differences in KOR trafficking are entirely due to differences in ECE2-mediated degradation of Dyn B. In the revised manuscript, we have provided new data and discussion on why peptide degradation cannot fully explain the differences in trafficking between Dyn A and Dyn B.

The reviewer made this assumption based on two arguments. One, that the surface recovery rates of MOR, a different GPCR than the one we study here, is regulated by ECE2, and two, that ECE2 differentially processes Dyn A and Dyn B in the system used here. We feel that these data are not sufficient to make such a strong assumption, for several reasons.

First, relevant to the reviewer’s point about our previous data, we had not previously compared Dyn A and Dyn B-mediated receptor sorting into specific intracellular compartments and their recycling to the cell surface. Our use of advanced high-resolution imaging experiments in this study to carefully study KOR trafficking and signaling provides for a robust signal and allows for the careful comparison between Dyn A and Dyn B in KOR recycling.

Second, it is not accurate to equate the rates of surface recovery to differential location-based signaling. We have known now for over a decade that the rates of GPCR recycling can be regulated by signaling pathways, with functional consequences on signaling, without changing sorting, endosomal localization, or degradation on an ensemble scale (e.g., PMID: 16604070, PMID: 27226565, PMID: 25801029, PMID: 24003153). Therefore, the careful and direct characterization we show here is essential to understand how Dyn A and Dyn B differentially affect KOR localization and signaling.

Third, the correlation of ECE2 sensitivities and receptor trafficking is not established. ECE2 sensitivity for opioid peptides has been estimated using purified peptides and enzymes, and there is no evidence that the selectivity persists in vivo. In fact, most previous studies measured simply the effect of overexpressed ECE2. Further, the correlation is not obvious or direct for related peptide systems either. For example, we have found that BAM peptides, which also activate opioid receptors, drive less recycling of opioid receptors than Dyn B (data presented by Gupta, Gomes and Devi, INRC 2019) even though they are both ECE2 substrates (PMID: 12560336). We have presented data with BAM 22 peptide in Author response image 1. We hope that the reviewer will appreciate that a full and thorough characterization of the comparison of all of the BAM peptides and Dynorphins in the context of the complex roles of ECE2 is a separate study on its own, which is outside the scope of this manuscript that focuses on the physiological differences between two different endogenous opioids in cell lines and in neurons.

**Author response image 1. sa2fig1:** BAM22 drives noticeably less KOR recycling than DynB, even though they are both sensitive to ECE2 inhibition. (**A**) Recycling of HA-KOR to the cell surface after 30 min treatment with 100 nM Dyn B or BAM22 and washout for 0-120 min. Cell surface receptors were quantified by ELISA as described in Methods. Levels of cell surface receptors before agonist treatment were taken as 100% for each individual experiment. % recycled receptors were calculated by subtracting surface receptors at t = 0 (30 min internalization) from each recycling time point. (**B**) Cell surface receptors quantified after peptides were washed out for 120 min in media without or with 20 μm or the ECE2 inhibitor (S136492). The data represent mean ± SEM from three independent experiments carried out in triplicate.

Fourth, it has become increasingly clear that we cannot apply our understanding of one GPCR to the whole family. Many recent studies have highlighted how the mechanisms that regulate GPCRs and their functions diverge considerably between different GPCRs, even though the gross signaling characteristics are nearly identical. Relevant to the reviewer’s assumption, MOR and KOR recycle by distinct mechanisms. KOR uses a PDZ ligand on its C-term to recycle via an EBP-50-mediated mechanism (PMID: 12004055), while MOR uses a leucine-based sequence to recycle via an unknown mechanism (PMID: 12939277). Further, the recycling of MOR is regulated by different signaling pathways (PMID: 31575621) than those that regulate PDZ-mediated recycling of adrenergic receptors (PMID: 24003153). Therefore, it is not a given that the sorting of MOR and KOR will be regulated in the same manner.

Nevertheless, we appreciate that the reviewer’s suggestion to test whether the differences in KOR trafficking are entirely due to differences in ECE2-mediated degradation of Dyn B and not Dyn A. A key prediction of this idea is that inhibiting degradation of the peptides by ECE2 inhibitors would decrease the recycling of Dyn B-activated KOR, but have no effect on Dyn A-activated KOR. We have tested this prediction using ECE2 inhibitors and included the data in the revised manuscript. Contrary to the prediction, however, ECE2 inhibitors decreased KOR recycling after both Dyn A and Dyn B (Figure S2B). General protease inhibitors did not decrease the rates of KOR recycling after Dyn A or Dyn B (Figure S2A). We thank the reviewer for suggesting this experiment. In addition to including this data in the manuscript, we have added a description of the data to address why the differences in trafficking we see are unlikely to be due to differences in peptide processing.

Together, the new data support our conclusion that the differences we observe are not simply because of Dyn B degradation by ECE2, and suggests that the role of ECE2 in regulating sorting of KOR could be complex in vivo. While interesting, we feel that an in-depth characterization of how ECEs regulate peptide degradation in vivo is outside the focus of this study, which focuses on how two related endogenous opioid peptides, derived from the same precursor peptide, can differentially regulate KOR sorting and generate different spatial and temporal profiles of signaling.

4. The authors state "this is the first example of different physiological agonists driving spatial localization and trafficking of a GPCR" in light of the above comment, previous work from Bunnett et al., have shown how peptides with different endocytic enzyme sensitivity can indeed, localize GPCRs (e.g somatostatin receptor) in different compartments and elicit distinct signals (Padilla et al., J Cell Biol 2007; Roosterman et al., PNAS 2007; Zhao et al., JBC 2013 to name a few).

We were quite taken aback by this comment. We take previously published work very seriously, and we try to be as fair as possible when we describe that work.

We carefully searched through the papers the reviewer pointed out for an example where two physiological agonists drive different spatial localization and signaling of the same GPCR. But we could not find one. Padilla et al., 2007, compared different receptors. The manuscript shows that the recycling of CLR, whose ligand is degraded by ECE1, is sensitive to ECE inhibition, but that the recycling of angiotensin receptor or bradykinin receptor, whose ligands are not degraded by ECE, are not. Similarly, Roosterman et al., 2007, focus on how NK1 receptor recycling is sensitive to ECE1 inhibition. To the best of our knowledge, neither paper shows that spatial localization or location-based signaling of a given GPCR is regulated differentially by two different endogenous agonists. The closest experiment we could find was in Figure 2 in Zhao et al., JBC 2013. The main point of this figure is that “Agonists induce endocytosis of SSTR2A in myenteric neurons”. One panel in this figure shows that, when cells exposed to SST14 or the pro-peptide SST28 for 1 hour at 4°C are followed at 37°C and fixed, SSTR labeling at the plasma membrane and cytoplasm is similar at 30 min, but diverges after that. As far as we could decipher, receptor recycling, endosomal identity, or signaling were not tested in this manuscript.

Therefore, we respectfully request that the studies mentioned – of how the recycling of a receptor that binds ECE-sensitive ligands is sensitive to ECE inhibition – should not be conflated with our careful study of whether different endogenous opioids can drive different spatial localization and signaling fates of the same opioid receptor.

We have, however, modified the sentence to state the impact of our work more precisely. We have also cited the relevant paper to discuss the SSTR experiment in the revised manuscript. If the reviewer would point us to specific examples that show that subcellular localization and spatially restricted signaling of a given GPCR are regulated differentially by two different endogenous agonists, we will be more than happy to include a discussion of that work and modify the sentence further to match the current literature.

5. Support for endosomal signalling falls a bit short. For example, if indeed KOR signals from endosomes, the authors should use an inhibitor of receptor internalization and assess Nb39 recruitment and KOR signalling. Although this reviewer agrees that, since this experiment would still provide indirect evidence, it may not be necessary (though desirable) for publication.

We thank the reviewer for suggesting this experiment. As requested, we have included data showing that inhibiting receptor internalization with Dyngo-4a, a dynamin inhibitor, abolishes KOR localization in endosomes as well as Nb39 recruitment (Figure S3A-B).

Reviewer #2This manuscript demonstrates that two highly similar endogenous opioid agonists can give distinct opioid receptor trafficking and signaling fates. There are two key observations that are novel and intriguing: (1) two opioid peptides that are derived from the same precursor can distinctly modulate Kappa Opioid receptor (KOR) trafficking into two distinct pathways; Dynorphin A causes KOR trafficking to the late endosomes/lysosomes pathway whereas Dynorphin B promotes rapid recycling; (2) Dynorphin A activates Gi proteins on the late endosomes/lysosomes which leads to Gi-mediated cAMP inhibition from these compartments.The idea that GPCRs can activate G proteins at the late endosome/lysosomal compartments is fascinating and novel, however, the data presented here does not fully support their model that Dynorphin A activated Gi proteins on the late endosomes/lysosomes.

We are very happy that the reviewer found our study fascinating and novel. We thank the reviewer for the comments, and we have addressed them as follows.

1. There is a mismatch with the timing of receptor colocalization experiment (Figure 3B and C, 20 min Dynorphin A/B treatment) and the cAMP assay (Figure 3H, 5 min treatment). There needs to be direct evidence that KOR is localized on the late endosomes/lysosomes at 5 minutes post agonist stimulation, i.e. at the time that cAMP levels are measured. It is important to demonstrate that the sustained signaling inhibition by DynA comes from the late endosomes/lysosomes as opposed to early endosomes. A colocalization experiment with 5 min DynA stimulation followed by a 25min washout would be necessary to support their model.

We thank the reviewer for raising this important point. To clarify, the cAMP inhibition we observe at 5 min of agonist treatment is likely to be a result of signaling from the surface. The prolonged cAMP inhibition is measured after 5 minutes of agonist treatment and a 25 min washout, which is a total of 30 min. At this time, KOR is expected to be present late endosomes/lysosomes after Dyn A treatment. We apologize for not being clear, and we have clarified this in the revised manuscript.

As requested, we tested localization of KOR in Rab7 or Rab 11 endosomes under the same conditions (5 min agonist stimulation followed by a 25 min washout) in which we detected cAMP inhibition. Under these conditions, KOR is preferentially localized in Rab7 endosomes when activated by Dyn A, and in Rab 11 endosomes when activated by Dyn B, suggesting that Dyn A drives KOR to the late endosomal pathway even when agonist is washed out. We have added this new data as Supplemental Figure 3C-H.

In addition, we have also provided three-color live cell imaging data where we simultaneously localized the nanobody that recognizes active KOR with LysoTracker and KOR. As expected, all three markers colocalize in a subset of KOR endosomes, suggesting that at least a subset of the KOR in lysosomes is in the active conformation. We have included this data in the revised manuscript as Figure 3H and I.

2. What percentage of KORs are proteolytically degraded in the late endosomes/lysosomes at 20 min DynA stimulation?

At 20 min, although some of the receptors clearly reach the lysosome (Figure 3B-C, 3H), it is unlikely that there is significant degradation. This idea is supported by our immunoblots that show similar levels of KOR at 30 minutes after Dyn A and Dyn B (Figure 2I-J). This is also roughly consistent with previous studies on the difference in timing between GPCR localization in late endosomes/multivesicular bodies and receptor degradation. We realize this is an important point that we should address, and we have revised the manuscript to clarify these details.

3. Given that KOR trafficking to the late endosomes and lysosomes is mediate by ubiquitination (as shown here PMID: 18212250), does mutation of these ubiquitination sites (3 lysine residues on KOR C-terminus) block its trafficking and the sustained signaling from the late endosomes/lysosomes?

The reviewer raises an interesting topic that has been a subject of considerable debate in the opioid receptor field, and in the GPCR trafficking field in general. The mutation of the three lysine residues on the KOR C-terminus cause more residual KOR levels after 4 hours of Dyn A, suggesting that degradation/downregulation of KOR is reduced in these mutants, even though internalization is comparable (PMID: 18212250). For some opioid receptors, although ubiquitination might be required for involution and entry into the intralumenal vesicles, lysosomal localization is arguably independent of ubiquitination (e.g., PMID: 21106040, PMID: 22547407). Ubiquitination and/or lysine residues that interact with Ub-transferases could also affect downstream signaling, especially in the endosomes, by some GPCRs (e.g., reviewed in PMID: 30353650). Therefore, we feel that interpretation of results from the lysine mutant receptors will not be straightforward, whichever results we get from them. Nevertheless, we appreciate that this is an interesting point that needs to be addressed, and we have revised the manuscript to mention the complex roles of ubiquitination in receptor trafficking.

4. Is there any evidence for Gi protein localization on the late endosome/lysosomes?

This is another interesting point raised by the reviewer, as the majority of data on signaling from endosomes are on Gs-coupled or Gq-coupled receptors. However, Gi-coupled GPCRs, such as cannabinoid receptors or MOR can exist in the active conformation in endosomes (e.g., PMID: 18267983, PMID: 29754753), and internalization is required for sustained cAMP inhibition for the Class B S1P receptor (PMID: 24638168). These provide indirect evidence that Gi proteins might be present and active on endosomes. Unfortunately, directly testing whether Gi proteins are active on endosomes has been technically challenging. The main limitation has been the lack of conformation sensors for Gi proteins. We thank the reviewer for raising this important discussion, and we have discussed this point in the revised manuscript.

5. Additional functional readouts would also be helpful to support their model of Gi-mediated inhibition of cAMP response from late endosomes/lysosomes and not the plasma membrane or early endosomes. Perhaps mTOR activation (as authors have suggested in their discussion) could be used as a read out to show differences between DynA and B-mediated signaling?

We thank the reviewer for the suggestion. We used phosphorylation of the p70S6 kinase as a readout to test whether we could detect differences in mTOR signaling. However, under the conditions we tested, we did not see a significant increase in p70S6 phosphorylation upon activation of KOR with either dynorphin A or B. This could potentially be because there was high baseline p70S6 phosphorylation in the PC12 cells we were using. We have included the data as Supplemental Figure 4, and discussed the implications and considerations of the lack of a noticeable change in p70S6 phosphorylation. Since our data already suggest that there is an impact on cAMP signaling, we have still focused our discussion on the implications to cAMP signaling.

Reviewer #3This is an interesting idea and creative paper implicating a differential mechanism of intracellular trafficking and subsequently signaling that is triggered by different dynorphins binding to the kappa opioid receptor. However, there are some questions for the authors:

We thank the reviewer for the comments that the paper is interesting and creative, and for the insightful comments provided to improve the paper. We have addressed them as follows.

1. My reading is that some dynorphins are extremely rapidly degraded in serum and with these experiments performed in 15% Horse/FCS there is concern that some of the differential results could be explained by differential degradation. One hypothesis could be a differential frequency of receptor activation over time of a fast recycling receptor population. Can the authors convince me that this difference in trafficking and subsequent signaling is an intrinsic property of the peptide and not an exhaustion of peptide (would be DynB) over the 30min assay?

We agree this is an important point, and we apologize for not specifically explaining this point in the original manuscript. For the trafficking experiments, we directly compared results from experiments done with and without protease inhibitors. We saw no difference between the two conditions, possibly because we were using short time points, high enough concentrations, and dialyzed serum. We have included these data in the revised manuscript as Supplemental Figure 1A. The signaling experiments, which required longer incubations, were performed in the presence of protease inhibitors, consistent with previous studies. We have clarified this important point in the revised manuscript.

2. In Figure 2D, 2G and 2J at what time after addition peptides was this data obtained?

For measuring individual recycling events (2D and G), cells were treated with agonist for 5 minutes at 37°C. Receptor clustering was visualized using TIRF microscopy, and then a recycling movie was recorded at 10 Hz for 1 minute in TIRF. For 2J, we measured 2 time points, 30 min and 120 min after agonist addition. We apologize for overlooking these details, and we have included these details in the revised manuscript.

3. In Figure 2F the divergence of internalized receptor only occurs from time 20-30 mins which was difficult for me to understand since DynA should be resulting in lost surface receptor number. What confuses me is that in Fig2H the initial recycling induced by DynA17 is fast and slows down so I am wondering if a second hit is needed which feeds into my concern about peptide degradation in the media. Since released peptide would be pulsatile maybe in vivo DynA17 could act like DynB?

We apologize for not explaining the recycling experiment performed in 2F in more detail. The cells were imaged for a period of 2 minutes to collect baseline SpH fluorescence, which corresponds to the steady-state amount of KOR on the cell surface. After this period, cells were imaged for 15 min after Dyn A or Dyn B was added. In this period, because internalization is the predominant factor affecting surface levels, we see a loss in fluorescence as the receptors are internalized and SpH is quenched in the relatively acidic compartments. Because KOR internalization rates are not dramatically different between Dyn A and B, the fluorescence traces were not that different. The agonist was then washed out at this time (t=17), and cells were imaged in media containing antagonist. Because there is very little agonist-induced internalization after this point, the fluorescence change depends predominantly on reappearance of receptors via recycling. Therefore, if the main difference between Dyn A and Dyn B is in KOR recycling, we expect to see a divergence only in the late points of the trace. We thank the reviewer for carefully viewing the traces in 2F and 2H. We understand the interpretation that there might be fast and slow components to Dyn A induced recycling. While it certainly is possible, we are not comfortable making a strong conclusion on that, based on the sensitivity of the assays used and the variability between cells.

As mentioned in point#1, it is unlikely that this divergence in recycling is due to significant degradation of Dyn A. Nevertheless, it is an important point to discuss in light of the new data we provide, and we have explained this in detail in the revised manuscript.

4. The assays seem to be done with a single concentration of peptide – 1µM. Do the authors have data to show that at lower (or higher) concentrations than 1µM result in the same trafficking patterns, albeit to a lesser or greater extent. Also, for the cAMP inhibition what concentration gives max inhibition? For a binding affinity of 0.01nM in the cells and with high expression, the 1micromolar concentration seem high.

We thank the reviewer for raising this point. We used the 1µM dose based on dose-response measurements for cAMP signaling. Part of the dose-response data has been published (PMID: 32393639). However, we realize that this is a point that needs to be addressed in our system.

In the revised manuscript, we have included a dose-response for trafficking and signaling as Supplemental Figure 1. The reviewer is correct that we were using doses above saturation as far as the binding affinity goes, but the dose response depends also on receptor expression levels and availability of downstream components that determine the signaling and trafficking consequences. However, we were using saturating concentrations, and it is possible that this is what mitigates the potential degradation of small amounts of the peptides.

5. In Figure 2H 100% of receptors appear to be recycled after DynB however 25% of kappa colocalize in Rab7 in 3C so do these Rb 7 co-localized receptors recycle?

This is an interesting point, as it is certainly possible that some receptors from Rab7 endosomes can recycle. Current views, however, are more aligned with endosomes being overlapping populations as labeled by biochemical markers, especially by trafficking components like Rabs. Therefore, our characterization likely describes a spread of receptor distributions across these overlapping compartments. Moreover, the recycling of receptors in Figure 2H was quantitated using ELISA over 2 hours after agonist washout. The endosome colocalizations in 3C was measured after 20 min of agonist treatment. As we hope the reviewer would agree, it is difficult to directly compare data from these two experiments.

That said, we certainly did not mean to imply that all of Dyn B-activated KOR is recycled and that all Dyn A-activated KOR is degraded. Current data on trafficking support a more dynamic and flexible model for receptor sorting, where a fraction of the receptors is recycled while a fraction is degraded from each endosome. Our results are consistent with this model. We feel that, because the receptor populations undergo many rounds of rapid iterative sorting as the endosome matures, a larger fraction is recycled back to the surface in the case of Dyn B at a steady state, while a larger fraction stays behind in the case of Dyn A. Importantly, this difference in steady state localization is enough to cause a difference in endosomal receptor activation and cAMP signaling, suggesting that small differences in steady state localization can cause relevant changes in signaling. We apologize for not making this important point clearer, and have clarified this in the Discussion in the revised manuscript.

6. Could some of the signaling differences be explained by continued activation of receptor as a consequence of peptide processing in the endocytosed vesical as opposed to different vesicles? I guess the continued signaling could also direct subsequent trafficking and this could be tested with a membrane permeable antagonist.

We thank the reviewer for raising this point. As we described in our response to reviewer#1, peptide processing by ECE proteases could contribute to the differences, but the data suggest that this is not a direct correlation or the main explanation for the differences we observe. We have addressed this point in the revised manuscript.

7. The impact statement" Co-released dynorphins, which signal similarly from the cell surface, can differentially localize GPCRs to specific subcellular compartments, and cause divergent receptor fates and distinct spatiotemporal patterns of signaling" could be misconstrued. If one of the pathways is dominant and blocks the other, then co-release may only have one signaling outcome. Have any dynorphin mix experiments been conducted? What might be anticipated?

We agree that the question of whether one peptide is dominant is an interesting one in the context of the paper, and we thank the reviewer for pointing this out. That said, assay sensitivity has remained a long-standing problem when trying these mixed experiments in the endogenous opioid system. We tried an equimolar mix of dynorphins A and B in our high-resolution imaging assay and in traditional recycling assays. We have included this data in Author response image 2. We feel that including the data in the manuscript takes away from the main point of the manuscript, as there could be many reasons for this result, but we will be happy to defer to the reviewer. To focus our discussion on the differences between the two dynorphins and reduce ambiguity, we have deleted that sentence and discussed the release of dynorphins A and B in the discussion in more depth.

**Author response image 2. sa2fig2:** Dyn B is dominant over Dyn A for KOR endocytic trafficking. (**A**) Quantitation of the number of exocytic events/μm2/min, as in Figure 2, in SpH-KOR PC12 cells co-treated with 1μM of Dyn A and 1μM of Dyn B, compared to either on its own. The co-treatment mimics Dyn B. (**B**) SpH-KOR PC12 cells were treated with 100nM Dyn A, Dyn B or a combination of both for 30 min. Peptides were washed out and cells incubated for 60 min in media without the agonist. Surface receptors were measured by ELISA as described in Methods. Percentage of recycled receptors were calculated by subtracting surface receptors at t = 0 (30 min internalization; 0% recycling) from each recycling time point. Co-treatmentcauses KOR recycling comparable to Dyn B alone. The data represent mean ± SEM from threindependent experiments carried out in triplicate.

8. It looks like details for the ELISA measurements in the methods section was missing. Were the ELISA measurements done with untagged KOR or SpH-KOR? One might worry about the effects of the N-terminal SpH tag on KOR trafficking, and it would be nice if the fluorescence SpH-KOR data were supported by ELISA for untagged KOR. (At least some of the data is immunostaining of FLAG-KOR, which probably introduces only minimal perturbation).

We apologize for not including the details of the ELISA experiments. The ELISA experiments were performed essentially as described previously (PMID: 24990314; PMID: 24847082). We have corrected this oversight and included these details in the revised manuscript.

The reviewer’s concern about the tag is a valid one, and one that we are very careful about. Nterminal tags are commonly used to study GPCR trafficking, and are usually preferred as they reduce interference with the cytoplasmic sequences and trafficking machinery. We have used two different tags to label the receptor, both on the N-terminus. The ELISA measurements were done using FLAGtagged KOR. The trafficking and microscopy experiments were done with both FLAG-tagged and SpH-tagged KOR. The signaling experiments were also performed with SpH-KOR and FLAG-KOR. The results are consistent between all these experiments, suggesting that the differences we observe are not due to the tag. We have clarified this point in the revised manuscript.

9. Dynorphin A17 is a very sticky peptide and difficult to wash out. Since we don't have a dose response it may require only very doses to have full activation for cAMP inhibition. It would be nice to be able to discount this as a potential for prolonged activation after washout.

The reviewer brings up a good point. Dyn A is less sticky in media or solutions containing 150mM NaCl, but we realize that this is a concern that should be addressed. In our case, we picked the doses we used based on dose-response curves that we have performed for cAMP signaling for these peptides. Also, experiments where we added both B and A, where the effect of B seems dominant, suggests that the results are not due to the stickiness of Dyn A.

We realize that it is important to explain the choice of our concentrations better, and we have done so in the revised manuscript.

We thank the reviewers for the careful analysis of the manuscript and for all their comments. We feel that the inclusion of new data and the extensive revision has substantially strengthened the manuscript. We hope that the reviewers find the revised version acceptable for publication.